# Potential for Microbial Cross Contamination of Laundry from Public Washing Machines

**Kelly Whitehead [1], Jake Eppinger [1], Vanita Srinivasan [1], M. Khalid Ijaz [1,*], Raymond W. Nims [2] and Julie McKinney [1]**

[1] Reckitt Benckiser LLC, Global Research and Development for Lysol and Dettol, Montvale, NJ 07645, USA
[2] RMC Pharmaceutical Solutions, Longmont, CO 80501, USA
[*] Correspondence: khalid.ijaz@reckitt.com; Tel.: +1-201-476-7707

**Abstract:** Although clothes washing machines remove dirt, microorganisms are not reliably removed by modern cold-water machine-washing practices. Microbial bioburden on clothing originates from the wearer's skin, the environment (indoor and outdoor), and the washing machine itself. While most clothing microbes are commensals, microbes causing odors and opportunistic pathogens may also be present. Understanding the extent of microbial transfer from washing machines to clothes may inform strategies for odor control and for mitigating the transmission of microbes through the laundering process. This study was designed to quantify and identify bacteria/fungi transferred from laundromat machines to sentinel cotton washcloths under standard cold-water conditions. Bacterial 16S rRNA and fungal ITS sequencing enabled identification of microorganisms in the washcloths following laundering. Total plate-based enumeration of viable microorganisms also was performed, using growth media appropriate for bacteria and fungi. Opportunistic human bacterial pathogens, including *Enterococcus faecium*, *Staphylococcus aureus*, *Klebsiella pneumoniae*, *Acinetobacter baumannii*, *Pseudomonas aeruginosa*, and *Enterobacter* spp., were recovered. The fungal bioburden was ~two-fold lower than the bacterial bioburden. Most sequences recovered were assigned to non-pathogenic fungi, such as those from genera *Malassezia* and *Ascomycota*. These results suggest that public washing machines represent a source of non-pathogenic and pathogenic microbial contamination of laundered garments.

**Keywords:** laundry; public clothes washing machines; microbial cross contamination; washing machine biofilm; infection control in laundering; opportunistic pathogens; infectious agents

## 1. Introduction

Household laundry is a routine part of modern-day life, with the purpose of maintaining personal hygiene and cleanliness. Over 80% of Americans own a washing machine, with the majority of remaining households utilizing public or communal machines for laundering clothing and other textiles [1]. While the laundered items may be visually clean following a domestic cold-water laundry cycle, microorganisms may not completely be removed or inactivated by such current washing practices [2–4]. The cleaning efficiency of a washing machine is attributed to the agitation caused by the washing drum/tumbler, the use of water and detergent at a given temperature, and an appropriate length of washing/rinsing time [4]. The trend towards designing increasing energy efficiency into the machine laundering process has resulted in a decrease in mean washing temperature from 63 °C to 46 °C in Germany over the past 40 years [4]. In the United States of America (USA), even cooler temperatures are typically used, with ~45% of households using a cold-water setting (typically 14 ± 4 °C) for over half of their loads [5]. This lowering of the wash temperature has resulted in reduction in efficacy for inactivating microorganisms [4,5]. Similarly, consumer preferences now trend towards use of bleach-free detergents, a practice that also has tended to reduce the microbicidal efficacy of the overall laundering process [2].

Finally, the longer washing cycles required to remove stains under these conditions provide ambient temperatures for longer periods of time, resulting in conditions favoring microbial diversity (i.e., higher temperatures such as 63 °C might be expected to limit persistence of heat-labile species) and the possibility of creating biofilms within the washing machine [4]. As a result, a persistent microbiome may remain on laundry items and on the water-contact surfaces of the washing machine itself following regular washing cycles and continued use [6]. That is, there may occur a continuous repopulation of the washing machine microbiome from the clothing being laundered, as well as the potential for contamination of the clothes being washed from the machine microbiome [3].

Microorganisms on clothing come from three main sources: (1) the wearer's skin; (2) the external environment (indoor, including public spaces in the home, healthcare settings, food service/hospitality settings, and contact with others or with pets; and outdoors, including soil, water, etc.); and (3) the washing machine itself and the water source used. The latter can be a considerable source of contamination of clothing, particularly if the machine is poorly maintained and not thoroughly cleaned/sanitized between uses. Among the various locations sampled in washing machines, the detergent drawer has been shown to harbor the most diverse bacterial populations [2,4]. It has been suggested that this contamination is sufficient (and persistent enough) to cause a shift in the textile microbiome from primary contaminants (e.g., skin bacteria) to bacteria residing in the washing machine itself [3]. While the majority of these microorganisms are likely skin-derived and environmental commensals, the growth or survival of pathogenic species also may be encouraged in the intermittently warm, moist, organic matter-rich environment that a washing machine provides [2–4].

The typical usage patterns in public laundromats mean that the washing machines are run relatively continuously, without long periods of non-use (in most cases not allowing time for sufficient drying of machine surfaces between uses). Under these conditions, the resulting environment might be particularly favorable for microorganisms to thrive. Warm, moist environments with multiple niche surfaces also represent environments ideal for the growth of a hard-to-eradicate bacterial biofilms [7]. Biofilms are symbiotic communities of bacteria that reside on surfaces within different settings, such as in hospitals (medical devices), on industrial equipment, on household surfaces (piping, toilet bowls, and showerheads), and in appliances (refrigerators and washing machines) [8]. Microorganisms comprising biofilms are notably more tolerant to antibiotics, antiseptics, desiccation, and other environmental insults, and are an important cause of recurrent human infections [9]. Biofilms have long been implicated in washing machine malodor [6]. Gattlen et al. [8] reported an increased likelihood of biofilm formation on washing machine plastic filters, rubber tubes, and metal parts of the outer drum (locations in relatively continuous contact with water), although the inner drum surfaces have failed to yield biofilm microorganisms. These biofilms have been composed of 94 different microorganisms, of which about a third were considered potential human pathogens, including *Pseudomonas aeruginosa*, *Citrobacter freundii*, *Serratia marcescens*, and *Klebsiella pneumoniae* [8]. Additional studies [2–4] also have demonstrated the presence of bacterial biofilms in washing machines, but questions remain as to the risk the biofilm microbes pose to machine users or wearers of the laundered items.

Due to the potential for biofilm formation within a typical unsanitized public washing machine, it is possible that microbial exchange between the washing machine and the textiles being laundered might occur during routine use. This study was undertaken to determine the quantities and types of bacteria/fungi transferred from public washing machines to sterile sentinel washcloths during a single wash cycle, under standard cold-water laundering conditions. We used 16S rRNA and ITS sequencing to identify bacterial and fungal microorganisms on the sentinel washcloths, and these microorganisms included several biofilm-forming human pathogens. This study suggests that unsanitized public washing machines can represent a potential transfer (cross-contamination) source for pathogens and microbes capable of causing infectious disease and clothing malodor.

## 2. Materials and Methods

### 2.1. Washing Machine Sampling

Eighteen identical cotton washcloths (Room Essentials, Target) were washed with Tide detergent and dried in a Whirlpool dryer (model WET3300XQ0). These washcloths were then autoclaved (Steris Amsco Lab Autoclave 250, San Diego, CA, USA), using a standard dry autoclave cycle in sealed autoclave bags (Medline Sterilization Pouches (12″ × 15″)). The pre-cleaned and sterilized sentinel washcloths were then each washed individually (no other clothing items were laundered concurrently within the same loads) with Tide detergent (with no added bleach or laundry sanitizer) in 17 washing machines, in five public laundromats in Northern New Jersey and one laundromat in Rockland County, New York. This amounts to a single sampling event per washing machine evaluated. In each case, a cold-water cycle and small volume load setting was used in a variety of commercial front- and top-loading washing machines. The sentinel washcloths were retrieved from the washing machines using aseptic sampling measures and were brought back to the lab, while still moist, for processing. No more than 24 h elapsed between sampling and processing. The sentinel washcloths were stored overnight at 2–8 °C if needed. Note: out of 17 sentinel washcloths, only one set of samples, those for laundromat L1, were stored overnight at 2–8 °C and then processed.

### 2.2. Recovery of Microbial Load and Total Viable Counting Procedures

Sentinel washcloths were placed in a sterile Whirl Pac Stomacher bag containing 150 mL of 1× phosphate-buffered saline (Becton Dickinson GmbH, Heidelberg, Germany), and processed in a Seward Stomacher at 180–260 rpm for 10 min. Expressed liquid samples (one per sentinel washcloth) were plated for total viable bacteria on Tryptic Soy Agar and incubated at 35–37 °C for 48 h. To quantify the total viable fungi in samples, Hardy Malt Extract Agar (with chloramphenicol) or Potato Dextrose Agar (PDA, BD Difco, Sparks, MD, USA) was used and the inoculated plates were incubated at 29–31 °C for at least 10 days. The use of the different incubation temperatures for bacteria vs. fungi was predicated on the preferences of the two microorganism types for the higher or lower temperature, respectively. Our goal here was to isolate viable microorganisms, not to duplicate the conditions in which the washing microbiome microbiomes might exist.

### 2.3. Sample Processing for DNA/RNA Extraction

After plating, liquid samples (one per sentinel washcloth) processed via Stomacher were centrifuged at 10,000× *g* for 15 min in a Labnet Prism microcentrifuge. Multiple microcentrifuge runs, with subsequent pooling of the resulting pellets, were performed in order to compensate for significant sample volumes and to optimize the recovery of genetic material for identification. The final pooled pellets were resuspended in approximately 1 mL of DNA/RNA Shield solution (Zymo Research, Irvine, CA, USA) and sent to Zymo for sequence-based identification.

### 2.4. Sequence-Based Analysis of Sentinel Washcloth Microbial Flora

<u>DNA Extraction:</u> One of three different DNA extraction kits was used, depending on the sample type and sample volume. In most cases, the ZymoBIOMICS® DNA Miniprep Kit (Zymo) was used. For low biomass samples, the ZymoBIOMICS® DNA Microprep Kit (Zymo) was used, as it allows for a lower elution volume, resulting in more concentrated DNA samples. For larger sample volumes, the ZymoBIOMICS®-96 MagBead DNA Kit (Zymo) was used to extract DNA using an automated platform.

<u>Targeted Library Preparation:</u> Bacterial 16S ribosomal RNA gene sequencing was performed using the *Quick*-16S™ NGS Library Prep Kit (Zymo). The bacterial 16S primers amplify the V1-V2 and V3-V4 regions of the 16S rRNA gene. Fungal Internal Transcribed Spacer (ITS) gene-targeted sequencing was performed using the *Quick*-16S™ NGS Library Prep Kit, with custom ITS2 primers substituted for 16S primers.

The sequencing libraries were prepared in real-time PCR machines to control cycles and therefore limit PCR chimera formation. The final PCR products were quantified with qPCR fluorescence readings and were pooled together based on equal molarity. The final pooled libraries were cleaned up with the Select-a-Size DNA Clean & Concentrator™ (Zymo), then quantified with TapeStation® (Agilent Technologies, Santa Clara, CA, USA) and Qubit® (Thermo Fisher Scientific, Waltham, WA, USA).

Sequencing: Libraries were sequenced on an Illumina® MiSeq™ with a v3 reagent kit (600 cycles). The sequencing was performed with >10% PhiX bacteriophage spike-in.

Bioinformatics Analysis: Unique amplicon sequences were inferred from raw reads using the DADA2 pipeline [10]. Chimeric sequences were also removed with DADA2. Taxonomy assignment was performed using Uclust from QIIME v.1.9.1 (11) with the Zymo Research Database, a 16S database that is internally designed and curated as reference. Composition visualization, alpha-diversity, and beta-diversity analyses were performed with QIIME v.1.9.1 [11]. If applicable, taxonomic assignments that had significant abundance among different groups were identified by LEfSe using default settings [12]. Other analyses, such as heatmaps, Taxa2SV_deomposer, and PCoA plots, were performed with internal scripts.

### 2.5. Statistics (Viable Counts)

It should be noted that many of the values obtained for viable bacterial or fungal counts were reported as "<x CFU/washcloth sample" or ">x CFU/washcloth sample". Such values are not suitable for statistical analysis. To allow statistical analysis, we truncated these values to "x CFU/washcloth sample". Summary statistics were generated using the statistical programming language, R (R Core Team 2020). The mean CFU/sample (bacterial or fungal) and standard deviation (SD) of the mean in logarithmic form are reported.

## 3. Results

The transfer of bacteria within the microbiome residing in the public washing machines or the influent water from a given washing machine to washed sentinel washcloths (previously sterilized; one per washing machine evaluated) was assessed by both viable plate counting and 16S rRNA sequencing (Figures 1 and 2).

The sentinel washcloth read depth of unique 16S rRNA sequences varied across laundromats, and unique sequences ranged from 8 to 238. (Figure 1). *Proteobacteria* were the most heavily represented phylum, followed by *Firmicutes*. *Pseudomonadales* were identified in abundance, including members of the families *Moraxellaceae* (genus *Acinetobacter*) and *Pseudomonadaceae* (genus *Pseudomonas*) (Figures 1 and 3A,B). Sentinel washcloths laundered in different washing machines within the same laundromat tended to be colonized with the same genera or exhibited similar levels of microbial diversity (Figure 1).

The three samples from Rockland County, NY were dominated by the environmental organism *Vibrio* spp. (Figure 4), almost to the exclusion of other genera. Relatively few members of the *Staphylococcaceae* (Figure 3C) and other Gram-positive bacteria were found to be transferred to the sentinel washcloths from the washing machines.

In addition to many non-pathogenic environmental microorganisms identified, several human pathogens were recovered from the sentinel washcloths following laundering (Table 1). All representative members of the ESKAPE pathogens [13], including *Enterococcus faecium*, *Staphylococcus aureus*, *Klebsiella pneumoniae*, *Acinetobacter baumannii*, *Pseudomonas aeruginosa*, and *Enterobacter* spp., were identified. Most were Gram-negative and known to be capable of biofilm formation (Figure 3).

Fungal ITS sequencing also identified a diverse eukaryotic biome in the influent water or public washing machines (Figure 5), with an average of 27 (range 15 to 52) unique sequences found transferred to the sentinel washcloths. Most sequences belonged to non-pathogenic fungi, such as those belonging to the genera *Malassezia* and *Ascomycota*.

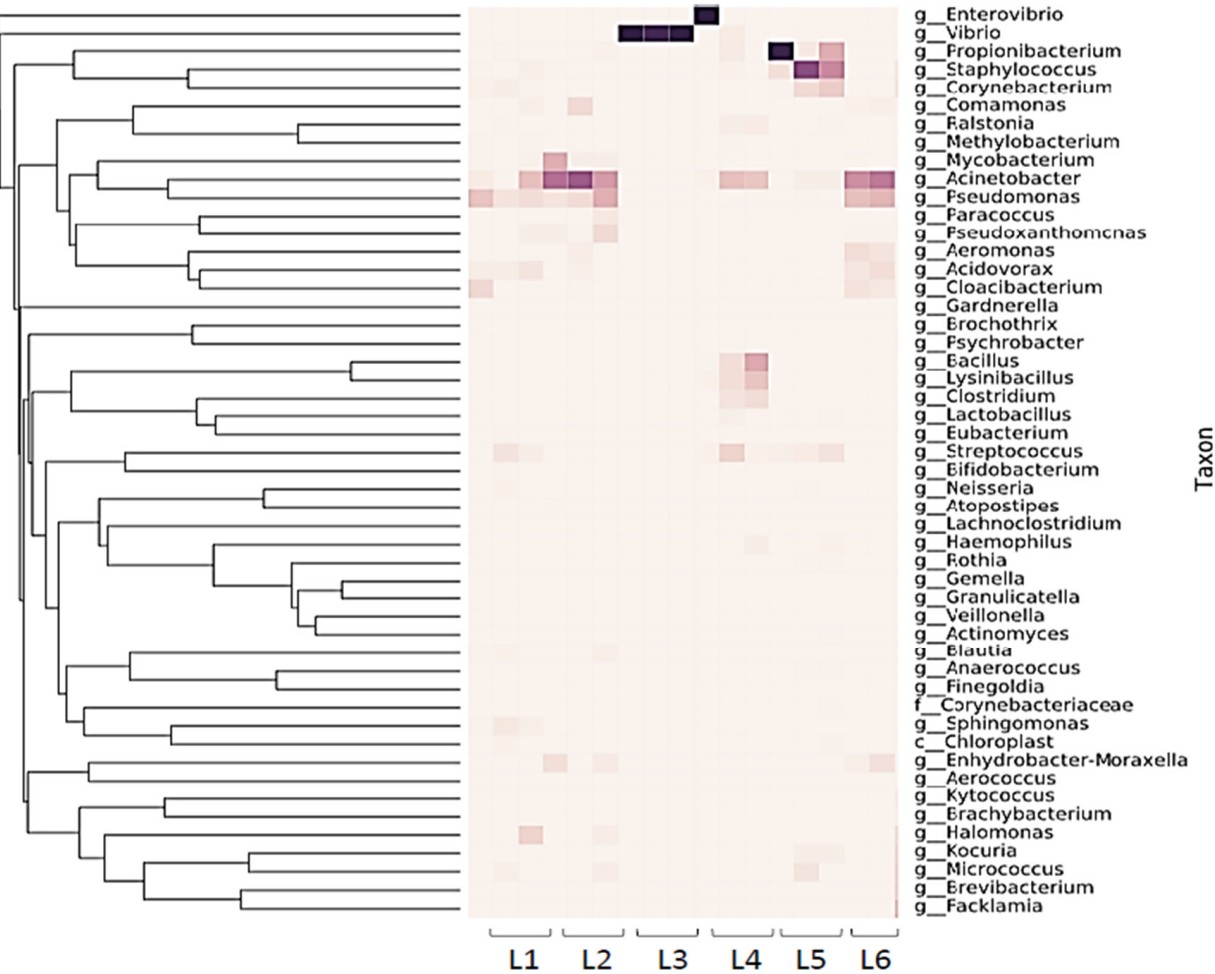

**Figure 1.** 16S rRNA sample heatmap showing diversity of 16S rRNA reads from sentinel washcloths laundered in 6 public laundromats (one washcloth per washing machine, with the laundromats designated as L1–L6).

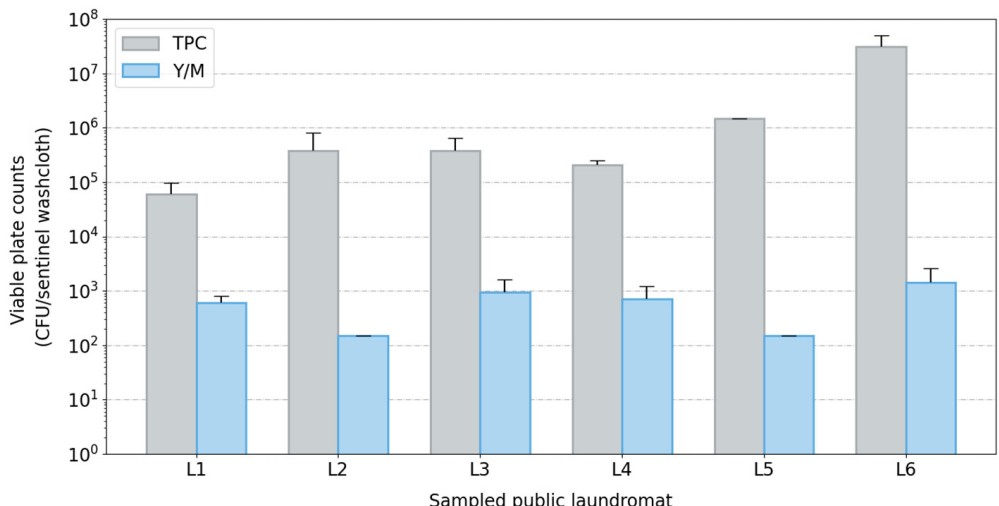

**Figure 2.** Bar plot of viable plate counts across sampled laundromats (L1–L6). Viable plate counts (bacterial and fungal) from extracted sentinel washcloths (one per washing machine) post-laundering using a cold-water cycle in public washing machines. Abbreviations used: CFU, colony-forming units; TPC, total plate counts; Y/M, yeast and mold counts.

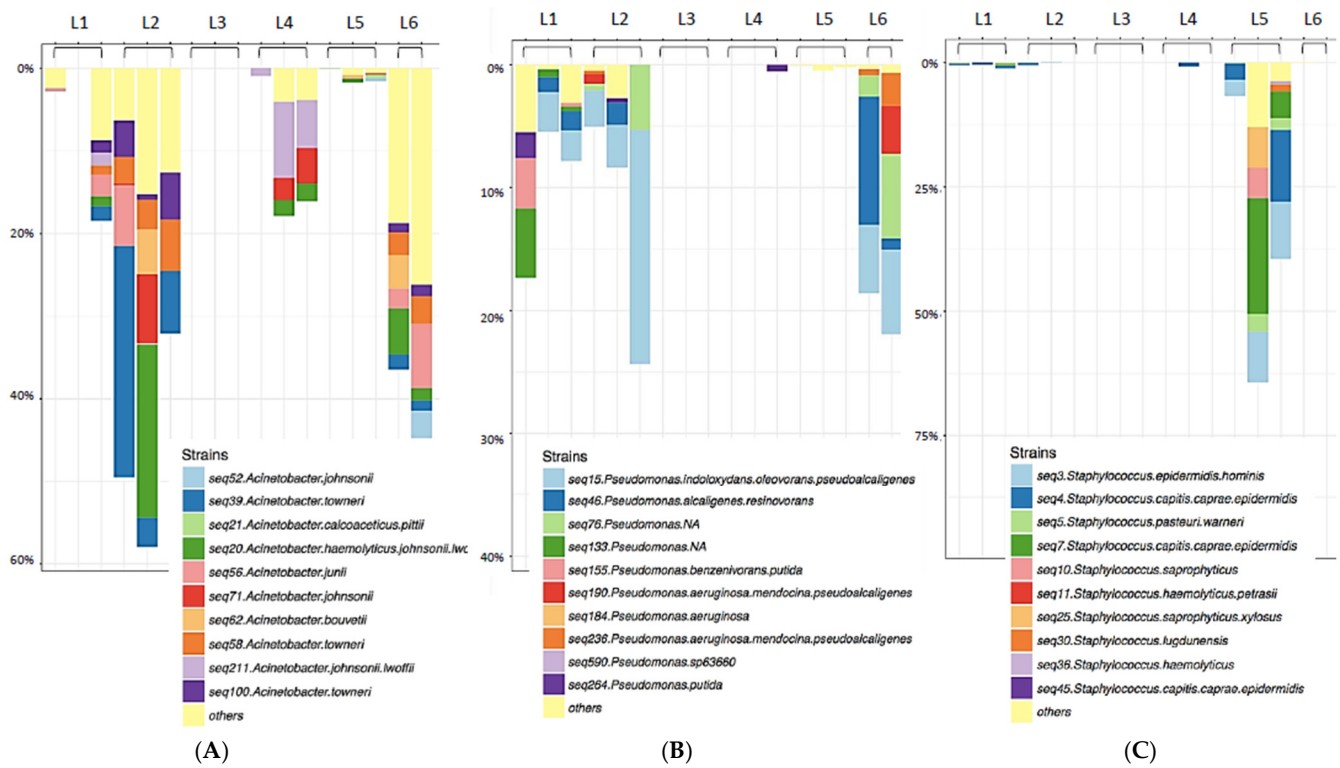

**Figure 3.** Genus-specific Taxa plots displaying *Acinetobacter* (**A**), *Pseudomonas* (**B**), or *Staphylococcus* (**C**) species diversity. Species diversity of key biofilm-forming nosocomial pathogens identified on sentinel washcloths (one per washing machine) after laundering in public washing machines.

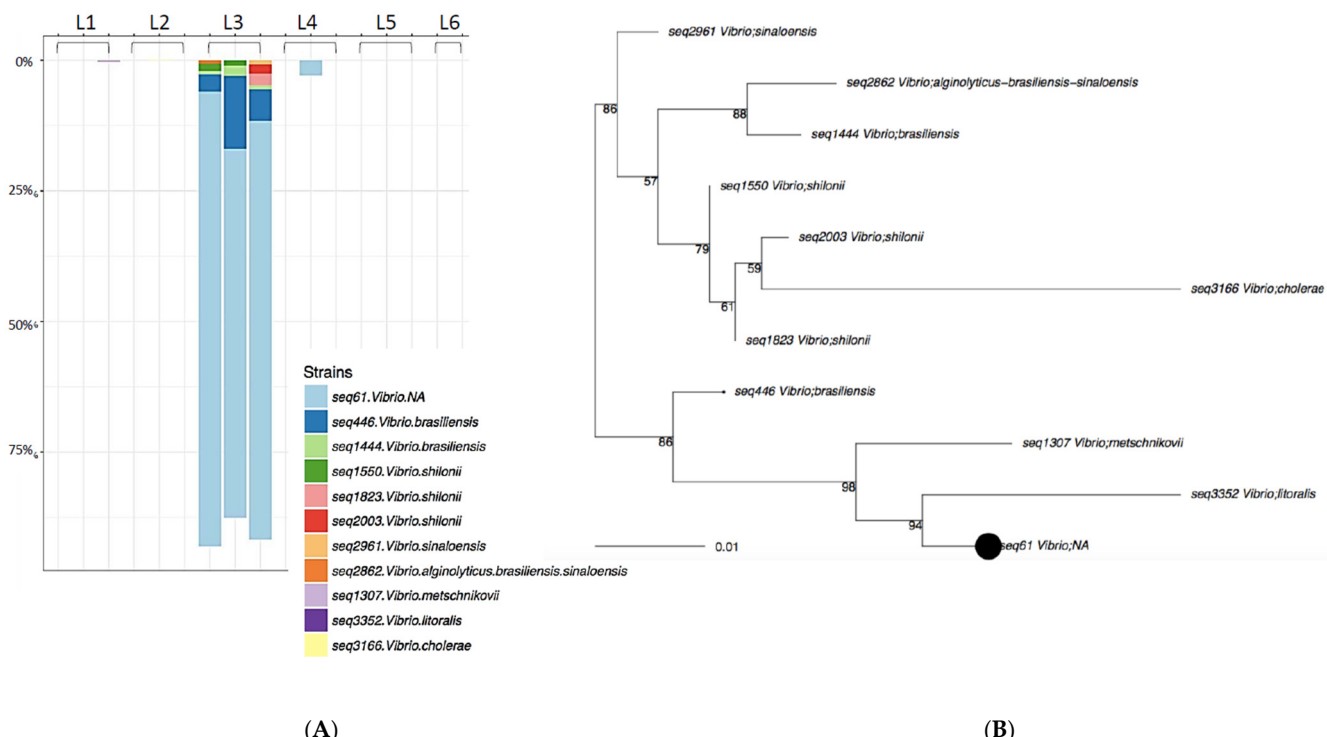

**Figure 4.** Laundromat L3 rRNA Taxa plot (**A**) and sequence diversity (**B**). *Vibrio* species were identified in high abundance on samples washed in Rockland County, NY (L3).

**Table 1.** Human Pathogens Identified by 16S rRNA Sequencing on Sentinel Washcloths after Laundering in Public Washing Machines.

| Pathogen | Pathologies | Biofilm Formation |
| --- | --- | --- |
| *Staphylococcus* spp. | Skin infections; more rarely, pneumonia, endocarditis, osteomyelitis | Yes [14] |
| *Enterococcus faecalis* | Urinary tract infections; more rarely, meningitis, endocarditis, sepsis | Yes [15] |
| *Haemophilus parainfluenzae* | Endocarditis, otitis media | Yes [16] |
| *Pseudomonas aeruginosa* | Wound infections, lung infections (cystic fibrosis, chronic obstructive pulmonary disease) | Yes [17] |
| *Acinetobacter baumanii* | Wound infections, sepsis, urinary tract infections | Yes [18] |
| *Klebsiella pneumoniae* | Wound infections, sepsis, urinary tract infections, pneumonia, meningitis | Yes [19] |
| *Stenotrophomonas maltophila* | Skin infections, endocarditis, meningitis, acute respiratory infections | Yes [20] |
| *Enterobacter* spp. | Skin and soft tissue infections, urinary tract infections, endocarditis | Yes [21] |

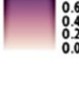

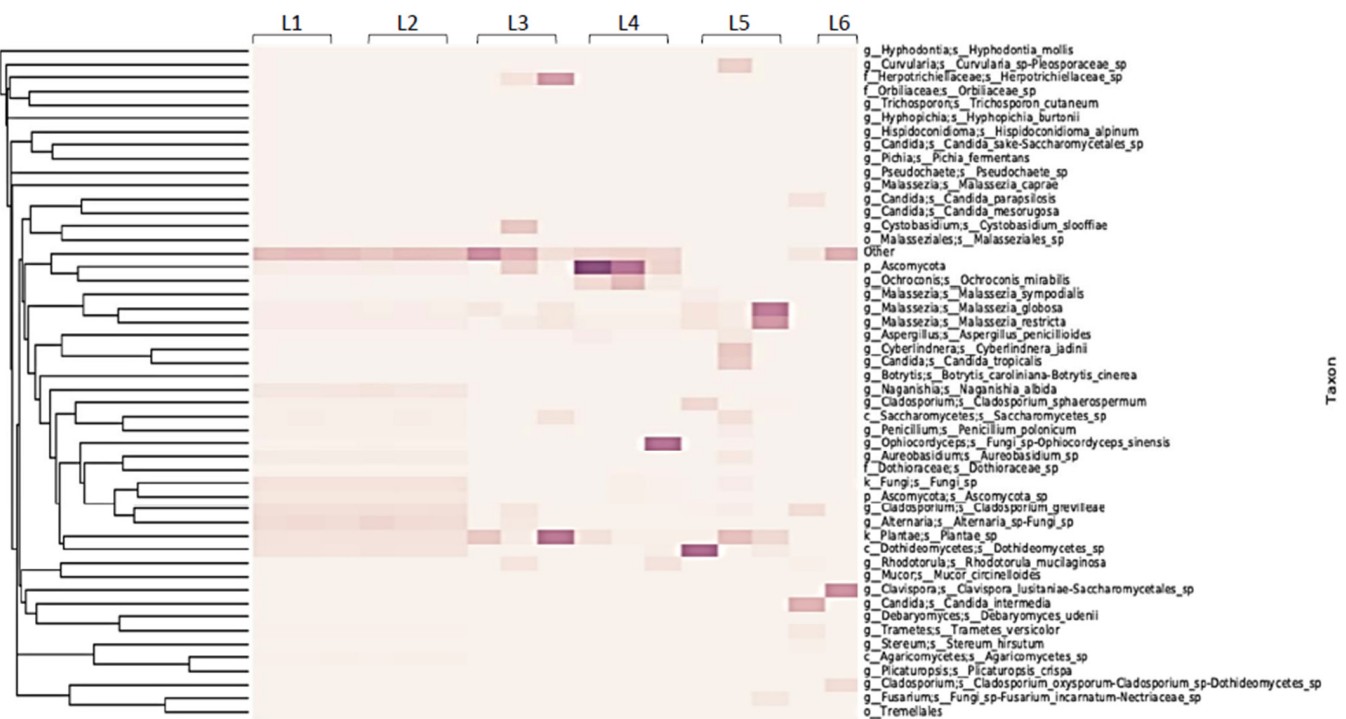

**Figure 5.** Heat map of fungal identifications from sentinel washcloths (one per washing machine) laundered in public washing machines.

Sequencing-based identification of bacteria has been criticized for its inability to distinguish viable cells from cellular debris or genomic material not associated with viable cells. To address this, we also isolated viable bacteria and fungi from sentinel washcloths using traditional plate-based techniques. Relatively high levels of viable bacteria and fungi were cultured from the sentinel washcloths following a single cold-water cycle. The average bacterial plate counts ranged from $10^4$ to $10^7$ CFU/washcloth, with a mean

value of $6.59 \times 10^6$ CFU (standard deviation: $1.01 \times 10^1$ CFU). Fungal counts were lower, with an average of $7.10 \times 10^2$ CFU (standard deviation: $4.5 \times 10^0$ CFU) (Figure 2). This result, combined with the results obtained with the sequencing-based (culture-independent) identification of microorganisms, suggests that a wide diversity of viable microorganisms (including pathogens) may be transferred to garments during laundering in public washing machines using a cold-water cycle.

## 4. Discussion

Modern clothes washing machines have been in use for approximately 100 years. These labor-saving devices have undergone significant re-design in the past quarter-century to adapt to changing consumer/societal preference for increased energy and water efficiency. Only an estimated 5% of household laundry in the USA is currently done at cycle temperatures above 60 °C [3], and detergents have been altered to include enzymes (and to exclude bleach) to permit the laundering of clothes under cold water (~40 °C conditions in Europe; ~16 °C conditions in the USA). While these modern clothes washing technologies may be efficient at removing stains, they may be less effective at sanitizing the washed clothes. In addition, the modern cycle parameters may permit the build-up of odor-causing microbes or pathogens in the machine itself, as well as on laundered articles [22].

The washing machine microbiome is established with contributions from the influent water, human skin flora derived from laundered articles, and bacteria biofilms residing in the washing machine itself [3,4,8]. Our goal in this work was to evaluate the level of microbial transfer from public washing machines to laundered articles, since such transferred microorganisms might include pathogens representing a potential for cross-contamination of clothes being laundered and subsequently, a source of infection to the wearer or laundry handler. Colony counts from sentinel washcloth samples on inoculated growth medium plates were used to quantify viable bacterial and fungal transfer from public washing machines to sentinel washcloths during a single cold-water cycle. In addition, 16S rRNA sequencing was used for identification of the transferred bacteria and ITS sequencing was used to identify the transferred fungi in order to assess the species diversity of the transferred microorganisms and to identify possible pathogens within the microorganisms transferred.

Not surprisingly, the results varied among the public washing machines evaluated. In most cases, a significant diversity in the species of microbes transferred was observed, with up to 238 unique species identified from a single washing machine. In a few cases, the microbes detected were limited to a single environmental genus (identified by sequencing as *Vibrio*). The lack of diversity in the latter samples was not due to reduced biomass in the machine, since total plate colony counts obtained from those machines (three washing machines evaluated in Rockland County, NY, USA) were comparable to those obtained from washing machines yielding more diverse genera. It is likely that *Vibrio* spp. were found in the washing machine influent water supply and/or existed as a biofilm within these washing machines. A limitation of the methodology of this study was that the microbiome of the influent water supply was not sampled and evaluated. This reflects the fact that the experimental work was conducted in public laundromats, and access to the water supplies to the machines was not possible. Additional studies will be needed to enable correlation of the detected microbiomes with the microbial content of the influent water supply.

Fungal colony counts were approximately 2 $\log_{10}$ lower than bacterial counts, and no pathogenic fungi were identified in the sentinel washcloths laundered in the six laundromats evaluated in this study. However, the transfer of approximately $10^3$ CFU/mL fungi to the sentinel washcloths confirmed that washing machines can be a source of fungal transfer and, though not observed in this study, the possibility for transfer of potential fungal pathogens cannot be ruled out.

A number of biofilm-forming microorganisms were identified during sampling of the sentinel washcloths (Figure 3). Biofilm formation within washing machines is a concern, not only due to the possible presence of malodor-associated microorganisms, but

also the possible presence of pathogenic [23] and drug-resistant microorganisms in such biofilms [16,21,24–26]. As is typical of biofilms, there is a likely persistence within the washing machines associated with the difficulty in removing/sanitizing the biofilms. Biofilms form long-lasting surface-associated communities in washing machines that resist environmental insults, such as detergents and desiccation [8]. As a result, there is always a chance of ongoing release of planktonic microbes from the biofilm during a wash cycle, while a nexus of microbial colonization in the biofilm is retained that is difficult to sanitize through use of laundry detergents that do not contain microbicidal actives. Unfortunately, certain human pathogens, such as *P. aeruginosa*, form biofilms efficiently, and the present results suggest that washing machine biofilms might represent a source of infection via pathogen shedding and transfer to washed garments. Opportunistic pathogens, such as *P. aeruginosa, A. baumanii*, and *S. aureus*, are able to colonize skin and wounds [20–25], and are members of the so-called ESKAPE pathogens, which have been singled out as threats to global health due to their high virulence and potential for acquiring multi-drug resistance [13]. Transfer of such pathogens from laundered clothes to open wounds may represent an especial concern for highly susceptible populations (e.g., immunosuppressed individuals [8] or those who suffer from conditions that pre-dispose to skin wounds, such as diabetes or psoriasis).

To our knowledge, this is the first time the possibility of transfer of microorganisms from public washing machines to clothing has been investigated. There have been similar studies published on the microbiome of domestic (home) washing machines [17,27,28], and also on the possibility of microbial exchange during the domestic clothes laundering process [29,30]. Jacksch et al. [4] evaluated 13 domestic washing machines, finding the greatest diversity of bacteria in the detergent drawer, followed by the sump and the door seal. The bacterial community identified contained members of the *Proteobacteria* (85%), *Actinobacteria* (5.3%), *Firmicutes* (3.0%), and *Acidobacteria* (1.1%) phyla, while the predominant genera included *Pseudomonas* (34.3%), *Acinetobacter* (17.4%), and *Enhydrobacter* (6.5%) [4]. In another paper by Jacksch et al. [27], four sampling sites (detergent drawer and detergent drawer chamber, and top and bottom rubber door seals) from 10 domestic washing machines were monitored for viable microorganisms. Of 212 isolates, 84% were identified to the genus level and 56% to the species level. The predominant bacterial genera were *Staphylococcus* and *Micrococcus*, and of these, 22 of 44 species were classified as opportunistic pathogens. Some of the species identified were considered biofilm producers (e.g., *Staphylococcus epidermidis, Micrococcus luteus, Bacillus cereus*, and *Pseudomonas* sp.) [27]. The total bioburdens obtained from the detergent drawer, detergent drawer chamber, and the bottom rubber door seal, were similar, while that from the top rubber door seal was ~1.5 $\log_{10}$ lower [26]. Gattlen et al. [8] evaluated 11 domestic washing machines, identifying 94 strains of bacteria or fungi, 30% of which were potential human pathogens (such as *Pseudomonas aeruginosa* and *Klebsiella pneumoniae*). The isolates were characterized by the authors as "typical environmental microorganisms inhabiting soil, water and the human body, including, among others, members of the Enterobacteriaceae and Pseudomonadaceae." [8] Biofilms were observed on metal, rubber, and polypropylene surfaces permanently in contact with water and not easily accessible for cleaning (e.g., plastic filters, metal parts of the outer drums, and rubber tubing). Total bioburden levels were similar among the 11 machines evaluated, though the bacterial compositions varied [8].

The possibility of bacterial exchange from domestic washing machines to laundered clothing was investigated by Callewaert et al. [3]. Samples of influent and effluent water were obtained from five domestic washing machines, and transfer from the washing machines to sentinel cotton T-shirts (not previously worn, but not pre-sterilized) was measured. Viable bacteria numbers were similar in the influent and effluent water samples, although a variety of biofilm-producing bacteria were found to be enriched in the effluent water samples. A variety of "typical skin- and clothes-related microbial species occurred in the cotton samples after laundering" [3]. The levels of certain bacteria were enriched following the laundering process. These included skin-related *Enhydrobacter, Acinetobacter*,

*Corynebacterium*, and *Staphylococcus* species, and biofilm-related *Pseudomonas* species [3]. The Schmithausen et al. paper [29] identified multi-drug resistant *Klebsiella oxytoca* (sequence type 201 and PFGE type 00531, a clone specific to this hospital and not previously isolated in Germany) in the detergent drawer and on the rubber door seal of a domestic washer-extractor machine that was used in the same ward to wash laundry for newborns. The isolates from the washing machine matched isolates taken from rectal or throat swabs and clothing of newborns in the ward, emphasizing the washing machine as a reservoir and fomite for the transmission of these multidrug-resistant bacteria [29].

In agreement with the results reported previously for domestic washing machines, we now demonstrate that public washing machines also can be a source of contamination of washed garments with a variety of bacteria and fungi, including potential pathogens. The items washed in public washing machines can include clothing from individuals who are symptomatic or asymptomatic carriers of infectious agents (bacteria, fungi, viruses, etc.) Once transferred to laundered articles, these pathogens may cross-contaminate the wastewater stream, the laundry-sorting table, the drying machine surfaces, and the wearer of the laundered clothing [3,29,30]. The results of this preliminary study need to be followed up in order to determine: (1) the correlation between the influent water microbiome and the species transferred to sentinel fabrics during the wash and rinse cycles; (2) the potential efficacy of laundry sanitizers on both biofilm presence on washing machine surfaces and the diversity and infectivity of microbes (including pathogens and malodor-producing bacteria) transferred to sentinel fabrics during washing in public washing machines; (3) the potential for transfer of enteric viruses (e.g., adenoviruses, rotavirus, hepatitis A virus [31]), and parasitic ova and (oo)cysts during the use of public washing machines; (4) the potential cross-contamination of sentinel washcloths from a pathogen-contaminated sorting table; (5) the role of soil bacteria in competitively eliminating biofilm-forming microbes from washing machine surfaces; (6) the types of microbes that are transferred to clothing articles during wear by apparently healthy male and female volunteers, including individuals carrying infectious agents, and that, therefore, could end up contaminating a washing machine; (7) the potential of bacteria containing antibiotic-resistance genes to contribute to the washing machine microbiomes and to transfer to washed clothing; and (8) the potential for allergens to transfer from the washing machines to laundered clothing. In addition, in this preliminary study, we were not able to explore the possible differences in microorganism exchange (quantity of microorganisms exchanged or types of microorganisms transferred) for textile types other than cotton. We acknowledge this as a limitation of our study, and hope that in the future this topic may be addressed by ourselves or others.

In internal laboratory studies, which were conducted per a globally approved ASTM method (Standard Test Method for Evaluation of Laundry Sanitizers and Disinfectants) [32] the microbicidal efficacies of laundry sanitizers and disinfectants were evaluated. Three laundry detergents commonly employed in the United States were tested against *S. aureus* and *K. pneumoniae* in room temperature water ($20 \pm 2.3$ °C) with a 16.5 min contact time (K. Smith, unpublished data). None of the laundry detergents reduced test microorganism titer by $\geq 3 \log_{10}$ (the efficacy expected of a laundry sanitizing agent). The efficacies measured were $0.85 \log_{10}$ (85.89%) reduction of *S. aureus* and $\leq 1.67 \log_{10}$ ($\leq 97.85\%$) reduction of *K. pneumoniae* on the tested cotton swatches. Within the recovered wash water from this experiment, the reduction of the test microorganism titer was found to be $\leq 0.90 \log_{10}$ for *S. aureus* and $\leq 0.41$ for *K. pneumonia*. An investigation by Schages et al. [30] arrived at similar conclusions for the efficacy of regular laundry detergent (i.e., no microbicidal additives) in cold-water (30 °C, somewhat higher than tested in our internal study) cycles of 30 to 60 min. Rather weak inactivation of *S. aureus* and *Trichophyton mentagrophytes* was observed (~$2.2 \log_{10}$), and somewhat higher inactivation ($\leq 4 \log_{10}$) occurred for *Enterococcus hirae*, *Pseudomonas aeruginosa*, and *Candida albicans* [30].

It appears from the above that regular laundry detergents (i.e., those not formulated with microbicidal additives) may be minimally effective (depending on the contaminating microorganism), when used in cold-water wash and rinse cycles in reducing the titers of

bacteria on clothing and, in fact, may leave viable microorganisms behind in the wash and rinse water. One might add a laundry sanitizer to the wash or rinse cycles to mitigate such risk. What efficacy would be required of a laundry sanitizer? The United States Environmental Protection Agency (EPA) is responsible for registering and regulating products that make sanitizer and disinfection claims on consumer products. A 3-$\log_{10}$ (99.9%) reduction in target microbial infectious titer is required [33] in order for a product to be registered as a laundry sanitizer and to make claims around pathogen reduction efficacy. Other mitigation strategies for reducing microorganism load in public washing machines might include: (1) implementation by the laundromat owner of ozone sparging of the influent water [34–36]; (2) selection of hotter water ($\geq$60 °C) wash or rinse cycles [5]; or (3) use of laundry detergent with microbicidal additives [5,31].

**Author Contributions:** Conceptualization, K.W. and J.M.; Data curation, K.W.; Funding acquisition, K.W. and J.M.; Investigation, K.W., J.E., V.S., M.K.I. and R.W.N.; Methodology, K.W. and J.M.; Project administration, K.W.; Writing—review and editing, K.W., J.E., V.S., M.K.I., R.W.N. and J.M. All authors have read and agreed to the published version of the manuscript.

**Funding:** This work was supported and funded by Reckitt Benckiser LLC. No outside funding was obtained.

**Data Availability Statement:** Supporting data for this paper may be requested from the corresponding author.

**Acknowledgments:** The authors would like to thank Zymo Research Corporation for their input and expertise in molecular identification and analysis, as well as Maria Wojakowski, PhD for quantitative statistical analysis.

**Conflicts of Interest:** R.W.N. received a fee from Reckitt Benckiser for editing of the manuscript. All other authors report no conflict of interest relevant to this article.

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
