# Peer review of "Potential for Microbial Cross Contamination of Laundry from Public Washing Machines"

_2036-7481, doi:10.3390/microbiolres13040072_

Round 1

Reviewer 1 Report

1.The theme is very interesting.It would be of interest to determine differencies between public self washing and public specialised places without self washing and also in home washing machines and also on food washing devices,or hotel laundry

2.Another viable method for identifing bacteria is MALDI-TOF.Was it available?

3.References 3,1319,20,22,25,27and others are missing pages

4.It would be of interest to add some data about the possible presence of the acarians like Demodex and its endosymbionts like Bacillus Oleronius,Pumilus,Simple ,Cereus,Kroppensteddi,because you mentioned Bacillus species.Please add and refer those data.

5.It would be of interest to add if possible more data on what founded germ can be involved in skin pathologies ,skin pathologies of the people.s clothes to be washed

5.Also if possible ,to  debate the type of germs and the type of the material of the clothes or the anatomical part involved in the clothes transmision of the germ -ex towel,slip etc

Author Response

Responses to reviewer comments.

Reviewer 1.

  1. The theme is very interesting. It would be of interest to determine differences between public self washing and public specialised places without self washing and also in home washing machines and also on food washing devices, or hotel laundry.

Our response: Thank you for your comment and your interest. You bring up some other worthy evaluations, which may be considered for future projects. Of course, in our discussion we did mention that most data on this topic have been generated for home laundry machines, and we have discussed in our current manuscript our results relative to the results from the published home laundry studies. We have also conducted studies on home washing machines, which we intend to publish separately. To add to your list of interesting future projects, the evaluation of hospital/healthcare setting laundry services might be a topic for future work.

  1. Another viable method for identifying bacteria is MALDI-TOF. Was it available?

Our response: We elected to use the sequencing method instead of MALDI-TOF.

  1. References 3, 1319, 20, 22, 25. 27 and others are missing pages.

Our response: Thank you for pointing out the missing information. We have gone through all of the references to ensure that the article numbers for these online references have been included.

  1. It would be of interest to add some data about the possible presence of the acarians [acariforms] like Desmodex and its endosymbionts like Bacillus Oleronius, Pumilus, Simple, Cereus, Kroppensteddi, because you mentioned Bacillus species. Please add and refer those data.

Our response: Unfortunately, we only obtained reads for Bacillus cereus and Bacillus cereus/pseudomycoides/toyonensis (i.e., identification to the exact species level was not obtained in that case)). The other Bacillus species you mention as endosymbionts of acariforms were not identified. If these endosymbionts were present in the washing machine microbiomes evaluated, we should have detercted their genomic material using the methods applied.

  1. It would be of interest to add some data if possible more data on what founded germ can be involved in skin pathologies, skin pathologies of the peoples clothes to be washed.

Our response: We are not sure what is being asked for here. This is the subject of Table 2.

  1. Also if possible, to debate the type of germs and the type of material of the clothes or the anatomical part involved in the clothes transmission of the germ – ex towle, slip etc.

Our response: Thank you again for your thoughtful questions. This subject was out of scope for the present study, which looked only at the question of bacterial/fungal exchange between public washing machines and sentinel cotton washclothes. But we do understand the question and its relevance. In a separate study being prepared for publication, we have looked at the types of bacteria/fungi that might be expected from different types of worn clothing.

Reviewer 2 Report

In this manuscript, the authors assess the potential for microbial cross contamination of cotton washcloths from public washing machines. The identification and quantification of bacteria and fungi was performed by resorting to amplicon sequencing and plating of viable cells.

The authors recovered opportunistic human bacterial pathogens, such as Enterococcus faecium, Staphylococcus aureus, Klebsiella pneumoniae, Acinetobacter baumannii, Pseudomonas aeruginosa, and Enterobacter spp., as well as non-pathogenic fungi of the genera Malassezia and Ascomycota.

Therefore, the study suggests that public washing machines represent a source of non-pathogenic and pathogenic microbial contamination of laundered garments.

The study is of relative interest and in the overall, the format and structure required by the journal are respected.

Nevertheless, some aspects of the methodology and the way some Figures are presented may be improved. Having this in mind, the authors should address the following aspects:

General comments:

1# The authors used 35-37C to quantify live bacteria and 29-31C to quantify fungi (L125-127). Can you comment why did you choose different temperature ranges to grow members of the same microbiome (for instance, why not grow bacteria at 29-31C)? Could the 35-37C biased the profile of bacteria recovered, favoring some members of species often associated to pathogenesis?

2# The number of replicates used in the sampling and for sequencing should be clearly stated in the Materials and Methods section. Even though I got an hint of the number of sequencing replicates in the Results section, the manuscript would benefit of having this information mentioned properly.

3# Could the authors clarify what was the contribution of the LEfSe analysis, given that I did not see reference to it in the Results and Discussion?

Detailed and minor aspects:

4# L166

The first sentence of the “Statistical analysis” can be further clarified. Please make it clearer.

5# Figure 1

A color key is missing in the taxa plot and heatmap of Figure 1.

In the heatmap, I would suggest to transform the abundances e.g. into log, to highlight differences between samples, rather than displaying an heatmap with the majority of the cells with the same “almost-background” color. Also, I do not understand how the taxa plot is useful or adds unique information in the way it is presented. The information regarding the taxonomic level is also missing. Can you please comment?

6# L178

The legend of Figure 2 does not state the meaning of “TPC” or “Y/M”.

7# L295

Please correct “Firmacutes” to “Firmicutes”.

8# L294

Please italicize name of bacterial phyla.

9# L306

Please italicize “et al” and the review the document accordingly.

Author Response

Responses to reviewer comments.

Reviewer 2.

  1. The authors used 35-37⁰C to quantify live bacteria and 29-31⁰C to quantify fungi (L125-127). Can you comment why did you choose different temperature ranges to grow members of the same microbiome (for instance, why not grow bacteria at 29-31⁰C)? Could the 35-37⁰C biased the profile of bacteria recovered, favoring some members of species often associated to pathogenesis?.

Our response: This is a great question. Our purpose in using the two different temperature ranges was to favor the isolation of bacteria and fungi, recognizing that these microorganism types have different preferred temperature ranges for growth on relevant media (e.g., see USP <61> and <62, where the temperature for growing bacteria is suggested as 30 –35⁰C and for growing fungi is suggested as 20 –25⁰C). Our goal here was to isolate viable microorganisms, not to duplicate the conditions in which the microbiomes might exist. We have added this information to the methods section.

  1. The number of replicates used in the sampling and for sequencing should be clearly stated in the Materials and Methods section. Even though I got an hint of the number of sequencing replicates in the Results section, the manuscript would benefit of having this information mentioned properly.

Our response: The numbers of replicates for testing and sequencing have been clearly identified in the revised manuscript.

  1. Could the authors clarify what was the contribution of the LefSe analysis, given that I did not see reference to it in the Results and Discussion?

Our response: Group comparisons and LEfSe biomarker discovery were reflected in the 16S/ITS amplicon sequencing service reports from Zymo. Thus, the LEfSe analyses were part of the identification process.

  1. The first sentence of the “Statistical analysis” can be further clarified. Please make it clearer.

Our response: We have now attempted to make our meaning clearer.

  1. Figure 1. A color key is missing in the taxa plot and heatmap of Figure 1. In the heatmap, I would suggest to transform the abundances e.g. into log, to highlight differences between samples, rather than displaying an heatmap with the majority of the cells with the same “almost-background” color. Also, I do not understand how the taxa plot is useful or adds unique information in the way it is presented. The information regarding the taxonomic level is also missing. Can you please comment?

Our response: We agree with the reviewer that the taxa plot really does not add much information and will be difficult for the reader to interpret. We have therefore removed the taxa plot from this figure.

L178. The legend of Figure 2 does not state the meaning of “TPC” or “Y/M”.

Our response: Thank you for pointing this out. We have now clarified all abbreviations used in the figure legend.

  1. Please correct “Firmacutes” to “Firmicutes”.

Our response: Done, thank you.

  1. Please italicize name of bacterial phyla.

Our response: Done, thank you.

  1. Please italicize “et al” and the review the document accordingly.

Our response: Done, thank you.

Reviewer 3 Report

Major comments:

1)    This preliminary study has investigated the carry-over of microorganisms from public washing machines to sterile cotton swatches in a cold-wash cycle with non-bleach containing detergent. The quantity of transferred microorganisms from public washing machines to cotton swatches was determined via culturing (TSA, 36°C; PDA, 30°C), and the identification was done via genotypic analyses. The authors could show that

a.     The microbiomes in the public washing machines are related when standing in the same laundromat. This is a very interesting aspect. The microbiome could be impacted by the water quality, cleaning  measures of laundromat, community using the laundromat etc.

b.     High numbers of microorganisms are transferred from public washing machines to former sterile cotton swatches. The genotypic analysis revealed that also potentially pathogens are transferred from machine to the load.

2)    The study is consistent, and creative and is a good overall starting point for further investigations. The current and important literature is mentioned.

4)    General consideration: It is very valuable that the genotypic analysis/identification was combined with culturing. However, it’s a pity that culturing was not extended to selective agars (respectively microscopy for fungi) to see if some of the identified pathogens are viable.

5)    Introduction: Relevance of textile type might also be mentioned when transfer of microorganisms is discussed from washing machines to load pieces, because the selective adhesion / growth on different fibre types seems important.

6)    Method section: The general course of action was understandable.

7)    Data quality: Number of sampled washing machines is with n= 17 high enough. Data were analysed with different statistical tools. Diversity indices (between washing machines vs between laundromats) could be a valuable addition.

8)    Results and discussion:

a.     Figure 2: The microbial counts are only expressed as mean values per laundromat, provide the individual microbial counts per washing machine as supporting documents

b.     342ff::

                                          i.    In addition (would be interesting to investigate in the future):

1.     prevalence of antibiotic resistance genes in transferred microorganisms from public laundry machines to load pieces

2.     Transfer of allergens from public laundry machines to load pieces.

Author Response

Responses to reviewer comments.

Reviewer 3.

  1. General consideration: It is very valuable that the genotypic analysis/identification was combined with culturing. However, it’s a pity that culturing was not extended to selective agars (respectively microscopy for fungi) to see if some of the identified pathogens are viable.

Our response: Thank you for the comment. We appreciate your concern. It is an acknowledged weakness of the sequencing data that it allows identification by analysis of genomic material and does not allow one to conclude whether the identified genomic material is associated with viable or non-viable microrganisms. We were limited in this screening effort to the more broad-based methodologies used.

  1. Introduction: Relevance of textile type might also be mentioned when transfer of microorganisms is discussed from washing machines to load pieces, because the selective adhesion / growth on different fibre types seems important.

Our response: This comment has been raised by the other reviewers as well. This topic does deserve mention in the manuscript, but perhaps in the discussion section, and as one of the stated limitations of this study. As you can appreciate, as a preliminary study we were unable to explore more fully the different types of textiles that might be laundered.

  1. Data quality: Number of sampled washing machines is with n= 17 high enough. Data were analysed with different statistical tools. Diversity indices (between washing machines vs between laundromats) could be a valuable addition.?

Our response: Thank you for this comment. We feel that the heat map itself provides sufficient indication of species diversity.

  1. Figure 2: The microbial counts are only expressed as mean values per laundromat, provide the individual microbial counts per washing machine as supporting documents.

Our response: The within-laundromat variance in plate counts is indicated by the standard deviations around the mean values indicated by the error bars

Figure 1. 342ff:: In addition (would be interesting to investigate in the future):

  1. prevalence of antibiotic resistance genes in transferred microorganisms from public laundry machines to load pieces
  2. Transfer of allergens from public laundry machines to load pieces.

Our response: We genuinely appreciate the interest that this work has elicited in this reviewer and the others, and the thoughtfulness of the suggestions received. Of course, we are not able to promise that we can attack all of the deserving issues that have come to light, but hope that this publication may inspire other investigators to enter the topical area to assist in these investigations. In fact, we do mention in our discussion that the “Schmithausen et al. paper [29] identified multi-drug resistant Klebsiella oxytoca (sequence type 201 and PFGE type 00531, a clone specific to this hospital and not previously isolated in Germany) in the detergent drawer and on the rubber door seal of a domestic washer-extractor machine that was used in the same ward to wash laundry for the newborns. The isolates from the washing machine matched isolates taken from rectal or throat swabs and clothing of newborns in the ward, emphasizing the washing machine as a reservoir and fomite for the transmission of these multidrug-resistant bacteria.”